# Synergistic Effect of a Flavonoid-Rich Cocoa–Carob Blend and Metformin in Preserving Pancreatic Beta Cells in Zucker Diabetic Fatty Rats

**DOI:** 10.3390/nu16020273

**Published:** 2024-01-17

**Authors:** Paula Gallardo-Villanueva, Tamara Fernández-Marcelo, Laura Villamayor, Angela M. Valverde, Sonia Ramos, Elisa Fernández-Millán, María Angeles Martín

**Affiliations:** 1Departamento de Bioquímica y Biología Molecular, Facultad de Farmacia, Universidad Complutense de Madrid, 28040 Madrid, Spain; paulaisg@ucm.es (P.G.-V.); tafernan@ucm.es (T.F.-M.); 2CIBER de Diabetes y Enfermedades Metabólicas Asociadas (CIBERDEM), Instituto de Salud Carlos III (ISCIII), 28029 Madrid, Spain; lvillamayor@iib.uam.es (L.V.); avalverde@iib.uam.es (A.M.V.); s.ramos@ictan.csic.es (S.R.); 3Instituto de Investigaciones Biomedicas Sols-Morreale (IIB-CSIC), 28029 Madrid, Spain; 4Instituto de Ciencia y Tecnología de Alimentos y Nutrición (ICTAN-CSIC), 28040 Madrid, Spain

**Keywords:** type 2 diabetes, pancreatic beta cells, macrophages, oxidative stress, inflammation, flavonoids

## Abstract

The loss of functional beta-cell mass in diabetes is directly linked to the development of diabetic complications. Although dietary flavonoids have demonstrated antidiabetic properties, their potential effects on pancreatic beta-cell preservation and their synergistic benefits with antidiabetic drugs remain underexplored. We have developed a potential functional food enriched in flavonoids by combining cocoa powder and carob flour (CCB), which has shown antidiabetic effects. Here, we investigated the ability of the CCB, alone or in combination with metformin, to preserve pancreatic beta cells in an established diabetic context and their potential synergistic effect. Zucker diabetic fatty rats (ZDF) were fed a CCB-rich diet or a control diet, with or without metformin, for 12 weeks. Markers of pancreatic oxidative stress and inflammation, as well as relative beta-cell mass and beta-cell apoptosis, were analyzed. Results demonstrated that CCB feeding counteracted pancreatic oxidative stress by enhancing the antioxidant defense and reducing reactive oxygen species. Moreover, the CCB suppressed islet inflammation by preventing macrophage infiltration into islets and overproduction of pro-inflammatory cytokines, along with the inactivation of nuclear factor kappa B (NFκB). As a result, the CCB supplementation prevented beta-cell apoptosis and the loss of beta cells in ZDF diabetic animals. The observed additive effect when combining the CCB with metformin underscores its potential as an adjuvant therapy to delay the progression of type 2 diabetes.

## 1. Introduction

Type 2 diabetes (T2D) is a chronic metabolic disorder characterized by dysregulation of glucose homeostasis leading to hyperglycemia. This condition is associated with a gradual decline in the ability to produce and release insulin, frequently on the background of insulin resistance [1]. In this context, pancreatic beta cells, which are responsible for insulin secretion, play a central role in maintaining glucose homeostasis [2] but their progressive deterioration and loss, exacerbated by prolonged hyperglycemia, contribute significantly to the development of diabetes and its associated complications [3]. Therefore, the preservation of beta-cell function and mass is fundamental for controlling T2D, restoring proper glucose control, and improving metabolic regulation. 

Oxidative stress induced by prolonged hyperglycemia is one of the factors contributing to the destruction of beta cells in diabetes [4]. Moreover, the intrinsically low expression of antioxidant enzymes in pancreatic islets exacerbates the impact of oxidative stress, resulting in consequences for beta cells [5,6]. Similarly, pancreatic inflammation, frequently observed in individuals with obesity and/or T2D [7,8], also contributes to the progressive loss of beta-cell function. The increase in macrophage accumulation in the islets during T2D provides signals that primarily drive beta-cell hyperplasia, but also increase the production of inflammatory cytokines, further accelerating beta-cell damage [9]. However, there is limited evidence supporting the beneficial effects of existing antidiabetic therapies in augmenting intracellular antioxidant defenses or reducing inflammation to ameliorate beta-cell damage in T2D. In the context of T2D therapeutics, metformin, the primary glucose-lowering agent, has been suggested to be effective in maintaining beta-cell function. Its potential efficacy seems to be more pronounced when used alongside other pharmacological therapies [10,11]. Interestingly, an increasing body of literature suggests that combining metformin with natural bioactive compounds may provide enhanced protection against diabetes-related complications [12,13]. Nevertheless, the specific impact of metformin in combination with dietary natural bioactive molecules on the preservation of beta-cell mass remains poorly studied. 

Polyphenols are food-derived compounds that have attracted significant attention in recent years due to their demonstrated antioxidant, anti-inflammatory, and antidiabetic properties [14,15]. Cocoa is recognized as one of the most plentiful sources of dietary polyphenols, mainly flavonoids, known for their beneficial effects against T2D and its related complications [16]. In particular, cocoa flavonoids have been demonstrated to protect the integrity of pancreatic beta cells against oxidative damage both in vitro [17] and in vivo in young prediabetic rats [18]. Likewise, carob, a Mediterranean legume with a high flavonoid content, has also demonstrated positive effects in the context of T2D [19]. Accordingly, we have developed a potential functional food, rich in high-molecular-weight polyphenols, combining cocoa powder and carob flour [20]. Our recent findings have demonstrated the effectiveness of this cocoa–carob blend (CCB) in combating T2D and shown its cardioprotective effects [13]. These results were obtained using Zucker diabetic fatty (ZDF) rats, a well-recognized preclinical model of T2D. Within the frame of this previous study, herein, we have investigated whether the antidiabetic properties of the CCB extend its beneficial impact on pancreatic beta cells in an established diabetic milieu. Furthermore, we explored the potential additive effect of this dietary supplement when administered in combination with metformin. 

## 2. Materials and Methods

### 2.1. Cocoa–Carob Blend Diet 

The cocoa–carob blend (CCB) consisted of a mixture of cocoa powder generously provided by Idilia (Idilia S.L., Barcelona, Spain) and carob flour (sourced from Casa Ruiz Granel Selecto S.L., Madrid, Spain) in a 60:40 ratio. This product is notable for its high content of flavonoid-type polyphenols (16.7 g/100 g). More detailed information regarding the composition of the CCB is provided elsewhere [20]. The CCB-enriched diet (10%) was prepared by incorporating 100 g/kg of the CCB into the standard diet (AIN-93G diet), and both the control and CCB diets were adjusted to be isoenergetic. 

### 2.2. Animals and Experimental Design

This study is an extension of a prior investigation aimed at exploring the cardioprotective effects of CCB supplementation in Zucker diabetic fatty (ZDF) rats, a preclinical model of T2D. The metabolic characteristics and the pathophysiology of the ZDF rats are very similar to those of humans with T2D, which makes this model highly suitable for the evaluation of the consequences of a CCB-rich diet on pancreatic beta cells in a diabetic context. We have chosen to use this animal model because it is not possible to carry out such experiments in humans with T2D. Detailed descriptions of the animals and experimental designs have been previously documented [13]. Briefly, male Zucker diabetic fatty (ZDF) rats and their corresponding lean Zucker (ZL) controls were used. Animals were purchased from Charles River Laboratories (L’arbresle, France) at 11 weeks of age and were acclimated under standard controlled conditions for a week. The ZDF rats were randomly assorted in the following groups of rats: ZDF Group: Zucker diabetic fatty rats that received the standard diet; ZDF (M) Group: diabetic rats that received the standard diet and metformin (300 mg/kg/day); ZDF (CB) Group: diabetic rats that received a 10% CCB supplementation in their diet; ZDF (CB + M) Group: diabetic rats that received both a 10% CCB supplementation in their diet and metformin (300 mg/kg/day). The ZL non-diabetic group received the standard diet (ZL). Animals had ad libitum access to water and food throughout the 12-week study period. At 24 weeks of age, the animals were fasted overnight and then euthanized. Pancreases were removed and stored under different conditions depending on the analyses to be performed. A portion of the pancreas was fixed overnight in 4% p-formaldehyde (PFA) in 0.1 M phosphate buffer (pH 7.4) and embedded in paraffin for histological and immunohistochemical analyses. Another part of the pancreas was frozen at −80 °C for subsequent analyses.

All experiments were conducted in accordance with European standards and community legislation [21] and Spanish regulations [22] and with the approval of the Committee on Animal Care and Use for Experimental Purposes of the Community of Madrid (PROEX 079/19).

### 2.3. Biochemical Determinations

Blood glucose levels were measured using an Accounted Glucose Analyzer (LifeScan España, Madrid, Spain). To analyze serum insulin and HbA1c levels, commercial kits were employed (Rat Insulin, Mercodia, Uppsala, Sweden; HbA1c Kit Spinreact, BioAnalitica, Madrid, Spain). Fasting glycemia and insulinemia were employed to calculate the homeostatic model assessment indices for insulin resistance (HOMA-IR) and secretion (HOMA-B).

### 2.4. Pancreas Homogenates

Frozen pancreas samples were homogenized at a 1:10 (*w*/*v*) ratio in extraction buffer [18]. Subsequently, they were centrifuged at 14,000× *g* for 60 min, and the supernatants were collected. The protein content in the homogenates was determined using the Bio-Rad protein assay (Bio-Rad, Madrid, Spain), following the manufacturer’s instructions.

### 2.5. Determination of Pancreatic Insulin Content

The total insulin content of the pancreas was determined as previously described [18]. Glands were minced and disrupted ultrasonically in acid ethanol (1.5 mL of 12 M HCl/l00 mL ethanol) in a 10 mL/g pancreas ratio, extracted overnight at 4 °C and centrifuged. After that, insulin was determined in the supernatant using an ELISA kit (Rat Insulin, Mercodia, Uppsala, Sweden).

### 2.6. Reactive Oxygen Species (ROS) Determination

ROS were quantified using a fluorometric assay based on the oxidation of reduced dichlorofluorescein (DCFH) to oxidized dichlorofluorescein (DCF) (Sigma-Aldrich) by cellular oxidants, resulting in fluorescence emission [23]. Pancreatic homogenates (20 μg of protein) were diluted with Locke’s buffer and incubated with 5 mM DCFH for 30 min in the dark. Fluorescence was measured at an excitation wavelength of 485 nm and an emission wavelength of 530 nm.

### 2.7. Determination of Carbonyl Group Content

Protein oxidation in pancreatic homogenates was measured as the content of carbonyl groups present in the samples [23]. Each pancreas sample (500 μg of protein) was derivatized with 0.2% 2,4-dinitrophenylhydrazine (DNPH) in 2 M HCl, with a parallel blank treated only with 2 M HCl. Proteins were precipitated using 20% trichloroacetic acid and centrifuged for 5 min at 10,500 rpm. The precipitates were washed twice with ethyl acetate:ethanol (1:1, *v*/*v*) and resuspended in 6 M guanidine. Absorbance was measured at a wavelength of 360 nm, with each sample corrected using its respective blank. Results are expressed as nmol/mg of protein, using a coefficient of extinction of 22,000 nmol L^−1^ cm^−1^.

### 2.8. Determination of Glutathione (GSH) Levels

The concentration of GSH was assessed using a fluorometric assay based on the reaction of GSH with o-phthalaldehyde (Sigma-Aldrich) at pH 8.0 [23]. Pancreas samples (300 μg of protein) were mixed in 50 mM phosphate buffer at pH 7.0 and precipitated with trichloroacetic acid. They were then centrifuged at 1400 rpm for 30 min. Fluorescence was measured at 460 nm emission and 340 nm excitation in a microplate reader (Bio-Tek, Winooski, VT, USA). The results were interpolated on a GSH standard curve (5–1000 ng).

### 2.9. Determination of Glutathione Peroxidase (GPx) and Glutathione Reductase (GR) Activity

To determine the activity of the antioxidant enzymes GPx and GR, pancreas samples (400 μg of protein) were mixed in Tris buffer 0.25 M, sucrose 0.2 M, and DTT buffer 5 mM at pH 7.4, and then centrifuged at 3000× *g* for 15 min. GPx activity is based on the oxidation of GSH by GPx (Sigma-Aldrich) using tert-butylhydroperoxide (t-BOOH, Sigma-Aldrich, Madrid, Spain) as a substrate, coupled to the disappearance of nicotine adenine dinucleotide phosphate reduced salt (NADPH, Sigma Aldrich, Madrid, Spain) by GR. GR activity was analyzed by monitoring the oxidation of NADPH, which is used in the reduction of GSSG [23].

### 2.10. Western Blot Analysis

Pancreas samples were lysed at 4 °C in a lysis buffer. Afterward, supernatants were collected, and protein concentration was determined. These protein samples were then aliquoted and stored at −80 °C until they were utilized for Western blot analyses. SDS-polyacrylamide gel electrophoresis was employed to separate equal amounts of proteins, which were subsequently transferred onto polyvinylidene difluoride filters (Bio-Rad). Next, the membranes were probed with anti-TNF-α (sc-52746, Santa Cruz Biotechnology), anti-IL-6 (sc-57315, Santa Cruz Biotechnology, Quimigen, Madrid, Spain), anti-p65 nuclear factor kappa B (NFκB) (8242S, Cell Signaling, Madrid, Spain) and anti-phospho (Ser 536)-p65 NFκB (3033, Cell Signaling, Madrid, Spain), followed by incubation with peroxide-conjugated anti-rabbit (A6154, Sigma-Aldrich, Madrid, Spain) or anti-mouse (A4416, Sigma-Aldrich, Madrid, Spain) immunoglobulin. Protein bands were visualized using the SuperSignal™ West Pico PLUS Chemiluminescent Substrate (34580, ThermoFisher Scientific, Madrid, Spain) in an Imager2Imager CHEMI Premium (VWR). Band densitometry was quantified using ImageJ Software (v1.52a, National Institute of Health, Bethesda, MD, USA). To ensure the Western blot’s normalization, β-actin was employed.

### 2.11. Activity of Caspase-3

The activation of caspase-3 was assessed following a previously established protocol [18]. In brief, pancreatic tissues were lysed and 50 μg of protein for each condition were mixed with 20 mM HEPES (pH 7), 10% glycerol, 2 mM DTT, and 20 µM Ac-DEV-DAMC as the substrate. Enzymatic activity was determined by measuring fluorescence (excitation wavelength 380 nm and emission wavelength 440 nm).

### 2.12. Histological and Immunohistochemical Analysis

Pancreas samples embedded in paraffin (Panreac, Madrid, Spain) were cut into serial sections with a thickness of 5 μm using a microtome (Leica RM2125RT, Leica Biosystem, Madrid, Spain) and mounted on glass slides. For immunohistochemical analysis, serial sections of the samples were blocked with goat serum (S-1000, Vector Laboratories, Madrid, Spain) and incubated with a primary mouse antibody against insulin (I2018, Sigma-Aldrich) overnight at 4 °C. Subsequently, a secondary goat anti-mouse antibody conjugated with peroxidase (A4416, Sigma-Aldrich, Madrid, Spain) was added, and the sections were finally developed with DAB substrate (SK-4100, Palex Medical, Madrid, Spain) and counterstained with Harris hematoxylin (HHS32-1L). For fibrosis quantification, serial sections from samples of each condition were subjected to Masson’s trichrome staining (HT15-1KT, Sigma-Aldrich, Madrid, Spain), which stains fibrotic areas in blue. Images of the sections were acquired using a digital camera connected to a microscope (NIKON Eclipse 80i, Izasa, Barcelona, Spain). The percentage of beta-cell fractional area (brown) was analyzed using ImageJ v1.8 software and is expressed relative to the total pancreatic area measured from the sections of each condition. At least 30 sections per condition were analyzed. The percentage of fibrotic area was analyzed using the same software and is expressed as the percentage of collagen area (blue) relative to the total pancreatic area measured from the sections of each condition. At least 4 sections per animal and per group were analyzed.

Beta-cell apoptosis was estimated using the TUNEL method (ApopTag Peroxidase In Situ Apoptosis Detection Kit, Millipore, Madrid, Spain) coupled to insulin immunostaining, which was developed after incubation with the alkaline phosphatase secondary antibody with an alkaline substrate kit (SK-5100, Palex Medical, Madrid, Spain). The tissue was then counterstained with Harris hematoxylin. The beta-cell apoptosis rate is expressed as the percentage of apoptotic beta cells. At least 1500 beta cells were counted per pancreas. For cluster of differentiation 68-positive (CD68+) cell staining, rabbit anti-rat CD68 (ab125212, Abcam, Madrid, Spain) antibody was used coupled to insulin immunostaining as described for the TUNEL assay. The number of CD68+ cells was counted and is expressed per mm^2^ of total pancreatic surface area or as the number of CD68+ cells per islet distinguishing between islet peripheric localization, defined as CD68+ cells around the periphery of the islet, and intra-islet distribution, when CD68+ cells were located within the islet among the endocrine cells. Clusters of 6 or more beta cells were considered islets. A minimum of 60 islets were analyzed per animal.

### 2.13. Statistical Analysis

Data underwent statistical analysis using GraphPad Prism v8.2.1 software (GraphPad Software, Boston, MA, USA). The normality of the distribution was assessed through the Shapiro–Wilk test, and data were analyzed using one-way analysis of variance (ANOVA). Significant differences among the means of each condition were identified using the Tukey post-hoc test, with a significance level of 95% (*p* < 0.05). Results are expressed as mean ± standard error of the mean (SEM).

## 3. Results

### 3.1. Biochemical Characteristics of Diabetic Animals

At the study’s outset, ZDF animals exhibited a significantly higher body weight compared to ZL animals (340.3 ± 12.1 vs. 268.4 ± 9.8 g, respectively; *p* < 0.05), confirming their obese state. Additionally, at this initial time point, ZDF animals were diabetic, as evidenced by significantly elevated fasting glycemia when compared to the ZL group (182.2 ± 12.1 vs. 94.3 ± 8.1 mg/dL, respectively; *p* < 0.05).

Administration of the CCB, metformin, or both together over 12 weeks significantly reduced glucose, insulin, and glycosylated hemoglobin (HbA1c) levels, as well as decreased insulin resistance (HOMA-IR) and increased pancreatic function (HOMA-B) in ZDF animals (Appendix A). Notably, the combination of the CCB with metformin achieved values for both fasting glucose and HbA1c similar to those observed in the non-diabetic ZL group [13]. Altogether, these results indicated that the CCB diet effectively improved glucose homeostasis in ZDF rats, with the combined treatment of the CCB and metformin exhibiting even more robust protective effects than the other treatments.

### 3.2. Effect of the CCB and Metformin on Oxidative Stress Markers in the Pancreas of Diabetic Rats

In the context of T2D, oxidative stress induced by hyperglycemia plays a pivotal role in beta-cell dysfunction and apoptosis. Therefore, our initial focus was on investigating the oxidative state of the pancreas in diabetic animals. As illustrated in Figure 1a, a significant increase in ROS generation was observed in the pancreas of untreated diabetic rats (ZDF). However, treatment with metformin and the CCB diet led to a substantial reduction in ROS levels, effectively restoring them to levels comparable to those in non-diabetic ZL rats. Furthermore, we observed evidence of oxidative damage, as indicated by elevated protein carbonyl levels, in the pancreas of untreated diabetic rats (ZDF) but not in rats treated with metformin, the CCB, or the combination of both (Figure 1b). Likewise, the levels of the antioxidant molecule GSH were significantly reduced in ZDF rats compared to non-diabetic ZL rats, but treatment with metformin and the CCB significantly increased GSH levels, with a further improvement seen with the combination of both (Figure 1c). Regarding antioxidant enzymes, a notable increase in the activity of GPx was observed in all treated diabetic groups, with the most significant effect in ZDF rats fed the CCB-enriched diet (Figure 1d). The activity of the GR enzyme was significantly lower in ZDF rats, but not in other treated conditions, with similar levels to those found in the non-diabetic ZL group (Figure 1e). Collectively, these findings provide strong evidence that the CCB-enriched diet has the potential to enhance the activity of antioxidant defenses in the pancreas, particularly GPx, and reduce the oxidative stress induced by the diabetic condition.

### 3.3. Effect of the CCB and Metformin on Inflammatory and Apoptotic Markers in the Pancreas of Diabetic Rats

The pro-oxidant environment in the diabetic pancreas can also instigate inflammatory responses by activating the nuclear factor kappa B (NF-κB) signaling pathway, resulting in the upregulation of specific pro-inflammatory cytokines and pancreatic apoptosis. Consequently, our next step was to examine the impact of metformin and the CCB on markers of pancreatic inflammation (IL-6, TNF-α, and p-p65) and apoptosis (caspase-3 activity). Figure 2a,b, show a marked increase in the levels of IL-6 and TNF-α in the pancreas of non-treated ZDF rats. Both metformin and the CCB-rich diet were able to significantly decrease the levels of pro-inflammatory cytokines with a further significant reduction by their combination. Accordingly, treatment of ZDF rats with metformin and the CCB diet was also able to significantly diminish the activation of the NF-κB cascade by decreasing the levels of the phosphorylated p65 subunit (Figure 2c). Similarly, the activity of caspase-3 (Figure 2d), which is involved in the final steps of cell apoptosis, was significantly elevated in the pancreas of ZDF rats. However, CCB supplementation and metformin treatment partially prevented this increase. More importantly, the combination of the CCB with metformin totally avoided the increase in caspase-3 activity in the pancreas of diabetic animals. Taken together, these results provide strong evidence for the ability of the CCB to attenuate inflammation and apoptosis in the pancreas in a diabetic environment.

### 3.4. Effect of the CCB and Metformin on Islet-Infiltrating Macrophages in the Pancreas of Diabetic Rats

Macrophages are recognized as the primary immune cells involved in islet inflammation in both rodents and individuals with T2D. In fact, the infiltration of macrophages into islets during diabetes appears to mediate beta-cell destruction by secreting pro-inflammatory cytokines in close proximity to islet cells. For this reason, the presence of macrophages in the pancreas was investigated using CD68 staining (Figure 3a–c). In the lean condition, islet macrophages were mainly localized in the periphery, surrounding the islets. In contrast, the increased number of macrophages observed in non-treated ZDF pancreases (Figure 3b) was linked to a shift in the distribution of islet-associated macrophages with higher localization of CD68+ cells into the islets (Figure 3c,d). Metformin treatment of diabetic rats could not reduce the total number of pancreatic macrophages but substantially decreased the number of islets infiltrated with CD68+ cells. However, administration of the CCB-enriched diet, alone or in combination with metformin, normalized the number and distribution of macrophages. Collectively, these findings point out the capacity of the CCB to reduce islet macrophage infiltration and attenuate pancreatic inflammation in a diabetic milieu.

### 3.5. Effect of the CCB and Metformin on Beta-Cell Damage

To further confirm that the antioxidant and anti-inflammatory properties of the CCB contribute to protect pancreatic beta cells in diabetic rats, we conducted morphometrical analyses of islets in both lean and diabetic Zucker rats. As depicted in Figure 4a, islets in ZDF rats exhibited a loss of structural integrity compared to those in the control ZL rats. They showed increased extent and irregularity in their borders, with numerous extensions into the exocrine tissue and a heterogeneous staining pattern characterized by intense or weak insulin immunostaining of individual beta cells. Additionally, the sections analyzed in this condition revealed important peripancreatic fat accumulation and infiltrations of adipose cells into the pancreatic tissue, which could contribute to a local lipotoxicity. Moreover, the pancreas of ZDF rats exhibited a significantly higher percentage of fibrotic tissue compared to the healthy control group (Figure 4b). Treatment of diabetic rats with metformin alone failed to reverse the development of fibrosis. However, the administration of the CCB to ZDF rats for 12 weeks resulted in a significant reduction in the area of pancreatic fibrosis compared to untreated diabetic rats. Surprisingly, this positive effect on overall fibrosis was not found when combining the dietary treatment with metformin.

Consistent with these diabetes-induced structural changes, ZDF rats had significantly reduced beta-cell fractional area compared to non-diabetic ZL rats (Figure 4c). Nevertheless, all treatments tested resulted in a significant restoration of fractional beta cells compared to untreated diabetic rats (ZDF), reaching levels similar to those of the control group (ZL). Significantly, the CCB-enriched diet exerted the most pronounced effect. It is noteworthy that in the ZDF (CCB) and ZDF (CCB + M) groups, numerous clusters of two to six beta cells distributed throughout the pancreatic tissue were observed, indicating an active process of regeneration. This is consistent with the significantly reduced total insulin content found in the pancreas of ZDF rats that was completely avoided in all treatments (Figure 4d). These results highlight that CCB supplementation prevents beta-cell loss in ZDF rats.

To provide additional confirmation regarding the involvement of apoptosis in the loss of beta cells in Zucker diabetic animals, we specifically measured the apoptotic nuclei in beta cells in samples from the different groups by combining TUNEL staining and insulin detection. As illustrated in Figure 5a,b, positive TUNEL staining signals in the pancreatic islets of ZDF rats were prominently present, while these signals were scarcely detectable in the pancreatic islets of control ZL rats, pointing to a substantial increase in apoptotic beta cells in the pancreatic islets of ZDF diabetic animals. However, treatment with metformin or the CCB diet reduced the apoptotic effect, and the combined administration of both completely prevented it (Figure 5a,b). These results highlight the effectiveness of the CCB-enriched diet, alone or in combination with metformin, in preventing beta-cell apoptosis and thereby reducing the loss of beta cells in ZDF rats.

## 4. Discussion

Numerous studies emphasize the protective role of residual beta-cell function against the progression of diabetic complications, highlighting the significant value of treatments capable of sustaining beta-cell mass and function over time [24]. In the present study, we demonstrated that supplementation of diabetic animals with a promising flavonoid-rich functional food (CCB) significantly reduced pancreatic oxidative stress and inflammation, effectively preventing beta-cell loss and supporting better glycemic control. Importantly, its combination with metformin produced a superior effect, underscoring the potential advantages of combining natural bioactive compounds with antidiabetic drugs in diabetes treatment [25].

Markers of oxidative stress, such as increased ROS generation and diminished intracellular antioxidants, are consistent with pancreatic beta-cell damage in preclinical models and individuals with T2D [26,27,28]. Furthermore, the naturally low expression of antioxidant enzymes in pancreatic islets renders beta cells more susceptible to oxidative stress [29]. In line with this, our findings demonstrate that the CCB can mitigate ROS generation and oxidative damage in the pancreas of diabetic animals. Moreover, CCB supplementation enhances pancreatic antioxidant capacity in diabetic animals by boosting both enzymatic and non-enzymatic antioxidant defenses. Specifically, GSH levels exhibited a significant increase, with an additional benefit observed in combination with metformin. Similarly, CCB supplementation improved the activity of antioxidant enzymes, with a substantial increase in GPx activity compared to other treatments. Previous studies have demonstrated that polyphenols increase the expression and activity of these enzymes in various tissues [13,23,30,31] and in pancreatic beta cells [32,33]. Importantly, it has been established that overexpression of GPx-1 in beta cells of diabetic mice (db/db) preserved their mass and reversed the development of hyperglycemia [34]. In the same way, early treatment of diabetic ZDF rats with ebselen, a GPx mimetic agent, prevented beta-cell deterioration [35]. Taken together, these findings strongly indicate that a CCB-rich diet augments the activity of the antioxidant defenses in the pancreas, thereby reducing oxidative stress induced by the diabetic condition.

Additionally, ZDF diabetic animals showed pancreatic inflammation evidenced by increased levels of the inflammatory markers TNF-α, IL-6, and p-p65. Notably, CCB supplementation significantly attenuated the expression of all these pro-inflammatory mediators. Macrophages are the primary contributors to immune cell-mediated inflammation in islets [9]. Several studies have demonstrated an increased macrophage infiltration in T2D islets, often correlating with beta-cell dysfunction [36,37,38]. In our study, ZDF animals showed an increased number of macrophages primarily localized within the islets. Macrophages infiltrating diabetic islets express an M1-like pro-inflammatory phenotype, contributing to local overproduction of inflammatory cytokines and activation of the NF-κB signaling pathway, ultimately leading to beta-cell apoptosis and dysfunction [39]. Although metformin treatment did not reduce macrophage numbers, it did decrease their infiltration, thereby mitigating islet inflammation. This aligns with recent findings indicating metformin’s ability to inhibit islet inflammatory response and islet cell apoptosis in HFD/STZ-induced diabetic mice [40]. Importantly, both the number of macrophages and their infiltration into islets significantly decreased in the pancreas of CCB supplemented animals compared to untreated ZDF rats. In line with these data, other naturally derived compounds have been reported to attenuate inflammation in various tissues by reducing M1 polarization of macrophages in obesity and diabetes [41,42,43,44,45,46]. Notably, resveratrol, one of the most studied polyphenols, attenuated macrophage infiltration in pancreatic islets of T1D mice, thereby reducing beta-cell destruction [44]. More recently, it has been shown that the carotenoid lycopene may ameliorate hyperglycemia and dyslipidemia and attenuate beta-cell apoptosis by regulating the toll-like receptor 4 (TLR4)/NF-κB signaling pathway, both in islets of diabetic mice and in Min6 beta cells stimulated with the conditioned medium collected from RAW264.7 cells treated with glucose/palmitate [40]. Collectively these findings suggest that CCB treatment could prevent beta-cell loss in ZDF diabetic rats by inhibiting islet macrophage infiltration and intra-islet inflammation.

According to these results, the histological examination of the pancreas of ZDF animals revealed a reduction in the beta-cell fractional area compared to non-diabetic ZL controls. This decrease was accompanied by a lower cellular insulin content and structural changes in their islets, in alignment with observations from prior studies [45,46]. Notably, the CCB-enriched diet significantly prevented beta-cell loss, surpassing the efficacy of metformin treatment. Moreover, we observed that both the CCB and metformin treatments had a partial preventive effect on beta-cell apoptosis. These outcomes agree with prior studies demonstrating the anti-apoptotic properties of polyphenols as a significant mechanism for preventing beta-cell loss [5,32]. Unexpectedly, the beneficial effect of CCB supplementation on pancreatic fibrosis was not observed when the dietary treatment was combined with metformin; the mechanistic explanation for this requires further research. Altogether, the present results highlight the effectiveness of the CCB in preventing beta-cell loss in diabetic ZDF rats. Importantly, the combination of the CCB-rich diet with metformin was proven to be more effective in reducing beta-cell apoptosis, suggesting that their combined use may be therapeutically beneficial in reducing beta-cell loss during the progression of diabetes. However, in this study, we focused on the effect on this treatment on the whole pancreas, which is a limitation because it does not allow us to directly link the treatment to the functional changes in islets. Further studies with isolated islets could provide complementary information to elucidate the molecular mechanism involved in the beneficial effect of the CCB and metformin on beta cells.

It is noteworthy to mention that ZDF animals also exhibited increased insulin resistance, a condition that was significantly reduced by both CCB supplementation and metformin treatment (as indicated by HOMA-IR values). Although we did not specifically evaluate the effect of CCBs on liver function in ZDF rats, the existing literature suggests that cocoa flavanols may exert an insulin-like effect on human hepatic HepG2 cells under conditions of insulin resistance. This effect involves attenuation of the blockade of the insulin signaling cascade and modulation of glucose uptake and production [47]. Additionally, in pre-diabetic rats, a cocoa-enriched diet demonstrated the ability to alleviate hepatic insulin resistance by regulating key proteins of the insulin pathway and glucose metabolism in the liver [48]. Therefore, it cannot be excluded that the observed antidiabetic properties of CCB supplementation are partly due to the effects of flavanols on liver function and insulin resistance, hallmarks of T2D development and progression.

## 5. Conclusions

The present study demonstrates that chronic supplementation with a potential functional food rich in flavonoids (CCB) significantly prevents beta-cell apoptosis and the loss of functional beta cells in the islets of ZDF rats, ultimately improving glucose homeostasis in diabetic animals. The protective effect of the CCB appears to be partly mediated by its ability to enhance pancreatic antioxidant defenses, thereby neutralizing oxidative stress. Additionally, the action of the CCB in preventing macrophage infiltration into islets and the subsequent local production of pro-inflammatory factors significantly contributes to its beneficial effects on beta-cell apoptosis and dysfunction. It is interesting to remark on the observed additive effect when combining CCB treatment with metformin, which highlights its potential as an adjuvant therapy to delay the progression of diabetes.

## Figures and Tables

**Figure 1 nutrients-16-00273-f001:**
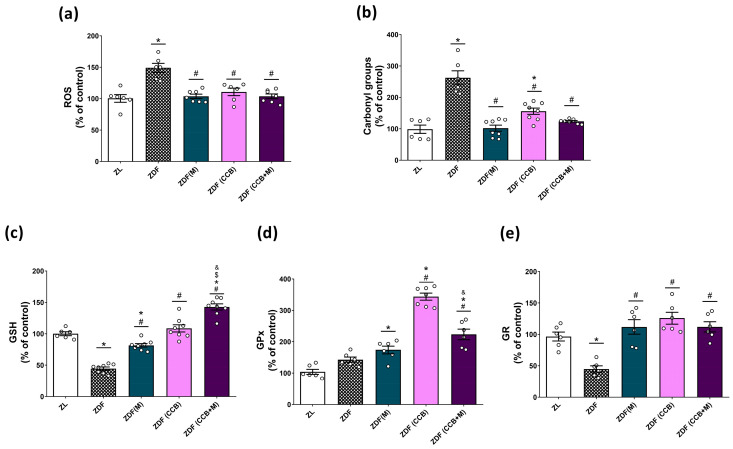
Effect of the CCB, metformin, and their combination on pancreatic oxidative stress. Percentage levels of ROS (**a**), carbonyl groups (**b**), GSH (**c**), GPx activity (**d**), and GR activity (**e**) relative to the control condition. Values are expressed as mean ± SEM of *n* = 6–8 animals. * *p* < 0.05 vs. ZL; # *p* < 0.05 vs. ZDF; $ *p* < 0.05 ZDF (CCB + M) vs. ZDF (M); & *p* < 0.05 ZDF (CCB + M) vs. ZDF (CCB). ZL: Zucker lean; ZDF: Zucker diabetic rats; ZDF (M): Zucker diabetic rats treated with metformin; ZDF (CCB): Zucker diabetic rats fed with a CCB-rich diet; ZDF (CCB + M): Zucker diabetic rats treated with metformin and fed with a CCB-rich diet.

**Figure 2 nutrients-16-00273-f002:**
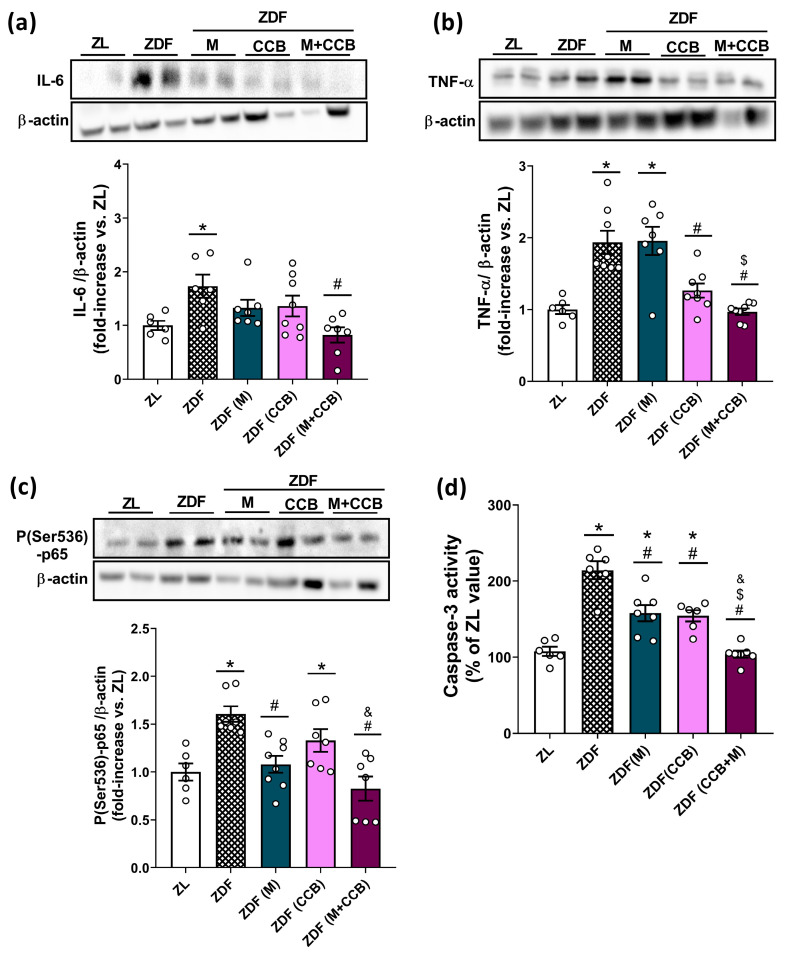
Effect of the CCB, metformin, and their combination on pancreatic inflammation and apoptosis. Representative Western blot analyses and levels of (**a**) IL-6, (**b**) TNF-α, and (**c**) P(Ser536)-p65 NF-κB in pancreatic tissues. Three independent experiments with two samples in each Western blot analysis were performed. (**d**) Fluorescent measurement of caspase-3 activity in pancreatic tissues. Values are expressed as mean ± SEM of *n* = 6–8 animals. * *p* < 0.05 vs. ZL; # *p* < 0.05 vs. ZDF; $ *p* < 0.05 ZDF (CCB + M) vs. ZDF (M); & *p* < 0.05 ZDF (CCB + M) vs. ZDF (CCB). ZL: Zucker lean; ZDF: Zucker diabetic rats; ZDF (M): Zucker diabetic rats treated with metformin; ZDF (CCB): Zucker diabetic rats fed with a CCB-rich diet; ZDF (CCB + M): Zucker diabetic rats treated with metformin and fed with a CCB-rich diet.

**Figure 3 nutrients-16-00273-f003:**
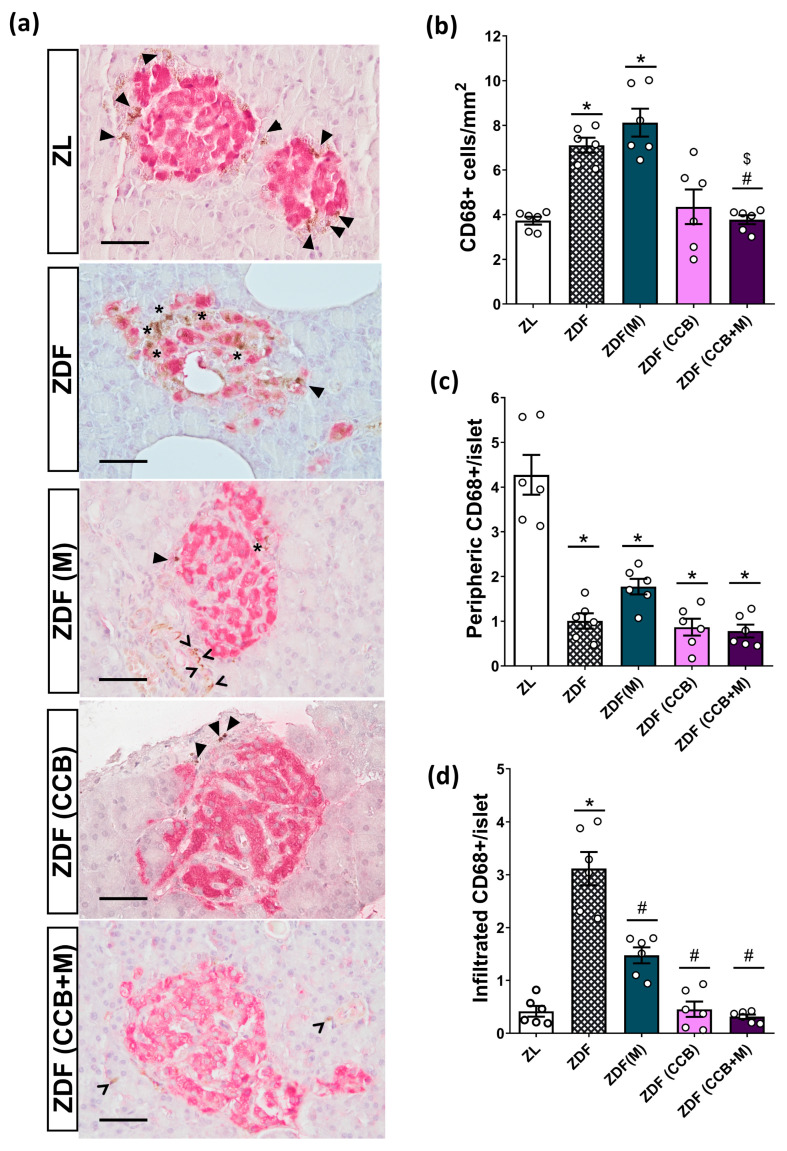
Effect of the CCB, metformin, and their combination on pancreatic macrophage infiltration. (**a**) Representative images of CD68+ immunohistochemical staining of islets (brown-stained). Scale bar = 50 μm. (**b**) Number of CD68+ cells expressed per mm^2^ of total pancreatic surface area. (**c**) Number of peripheric CD68+ cells per islet. (**d**) Number of infiltrated CD68+ cells per islet. In the images, intra-islet macrophages are marked with stars, peripheral macrophages are marked with a closed arrowhead, and exocrine macrophages are marked with an open arrowhead. In the graphs, values are expressed as mean ± SEM of *n* = 6–8 animals. * *p* < 0.05 vs. ZL; # *p* < 0.05 vs. ZDF; $ *p* < 0.05 ZDF (CCB + M) vs. ZDF (M). ZL: Zucker lean; ZDF: Zucker diabetic rats; ZDF (M): Zucker diabetic rats treated with metformin; ZDF (CCB): Zucker diabetic rats fed with a CCB-rich diet; ZDF (CCB + M): Zucker diabetic rats treated with metformin and fed with a CCB-rich diet.

**Figure 4 nutrients-16-00273-f004:**
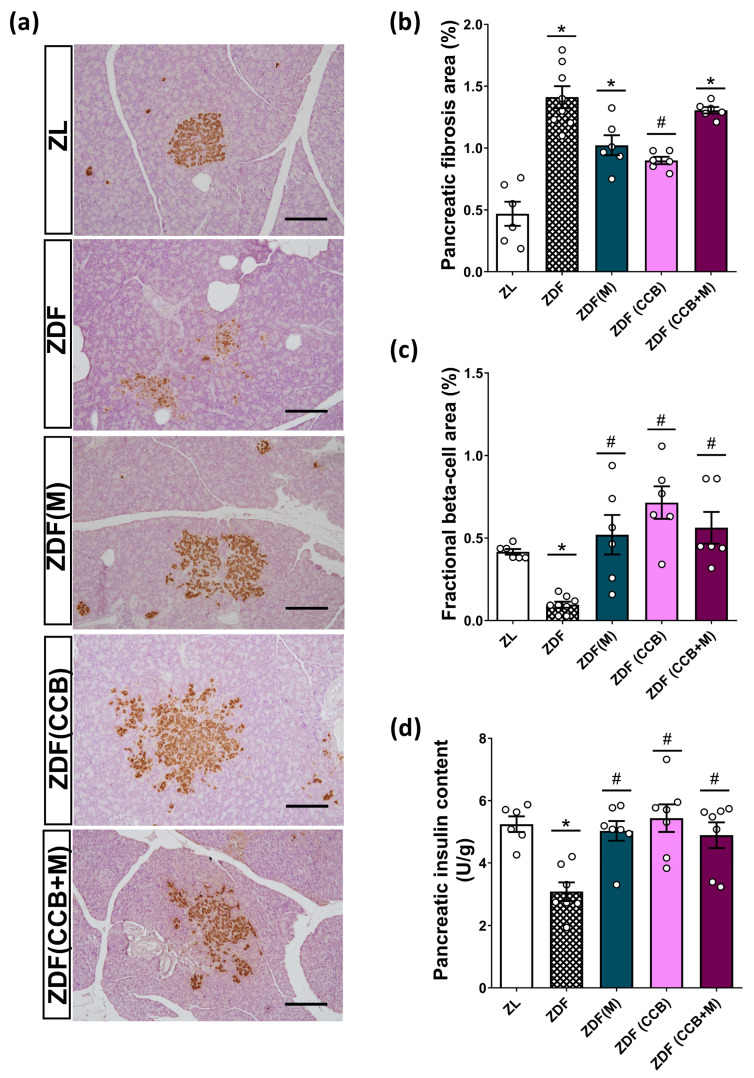
Effect of the CCB, metformin, and their combination on fractional beta cells. (**a**) Representative pancreatic tissue sections stained with insulin (brown) and hematoxylin (scale bars: 200 μm). (**b**) Quantification of fibrotic area expressed as the percentage of total pancreatic tissue. (**c**) Fractional beta-cell area (stained brown) expressed as relative to the total pancreatic area (20× magnifications). (**d**) Pancreatic insulin content. Values are expressed as mean ± SEM of *n* = 6–8 animals. * *p* < 0.05 vs. ZL; # *p* < 0.05 vs. ZDF. ZL: Zucker lean; ZDF: Zucker diabetic rats; ZDF (M): Zucker diabetic rats treated with metformin; ZDF (CCB): Zucker diabetic rats fed with a CCB-rich diet; ZDF (CCB + M): Zucker diabetic rats treated with metformin and fed with a CCB-rich diet.

**Figure 5 nutrients-16-00273-f005:**
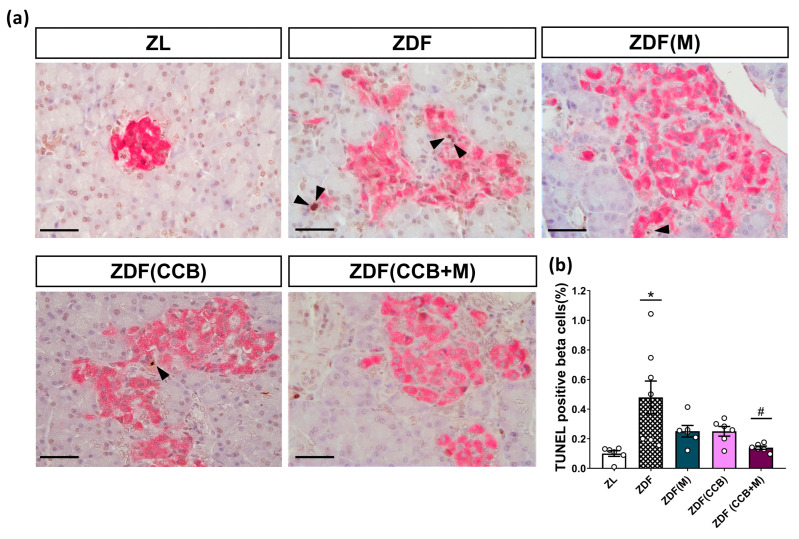
Effect of the CCB, metformin, and their combination on beta-cell apoptosis. (**a**) Representative immunohistochemical staining of terminal deoxynucleotidyl transferase dUTP nick end labeling-positive beta cells. Scale bar: 50 µm. Arrows indicate apoptotic nuclei (in brown) of pancreatic beta cells (insulin-positive, pink-stained cytoplasm). (**b**) Beta-cell apoptotic index expressed as the percentage of the total beta cells. Values are expressed as mean ± SEM of *n* = 6–8 animals. * *p* < 0.05 vs. ZL; # *p* < 0.05 vs. ZL: Zucker lean; ZDF: Zucker diabetic rats; ZDF (M): Zucker diabetic rats treated with metformin; ZDF (CCB): Zucker diabetic rats fed with a CCB-rich diet; ZDF (CCB + M): Zucker diabetic rats treated with metformin and fed with a CCB-rich diet.

## Data Availability

Data are available upon request to the authors. The data are not publicly available due to principle of confidentiality.

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
