# Peer review of "Synergistic Effect of a Flavonoid-Rich Cocoa–Carob Blend and Metformin in Preserving Pancreatic Beta Cells in Zucker Diabetic Fatty Rats"

_nutrients, 2024, doi:10.3390/nu16020273_

Round 1

Reviewer 1 Report

Comments and Suggestions for Authors

The manuscript is interesting but there are major flaws that significantly impair the data.

1. Insulin measurement from total pancreas measurements is inappropriate. The appropriate way to measure this is by isolating the islets and then measuring insulin content (with insulin secretion data). The exocrine pancreas will skew the results. 

2. Figure 2, 3, and 5 all show data from total pancreas lysates. These data, showing cascade 3 activity, ROS, carbonyl group, GSH, GPX activity, GR activity, and all western blots use total pancreas lysates. This issue inappropriate. Isolated islets should be used. With the islets being only 2% of the total pancreas weight, the 98% of exocrine pancreas could skew the results. Isolated islets must be used.

3. All the graphs should be changed from the plunger plots to bar graphs that show the individual data points. 

Author Response

The authors would like to thank the Reviewer for the careful reading of the manuscript and for the interesting suggestions and constructive comments, which have greatly improved the manuscript.
1. Insulin measurement from total pancreas measurements is inappropriate. The appropriate way to measure this is by isolating the islets and then measuring insulin content (with insulin secretion data). The exocrine pancreas will skew the results.

We appreciate the insightful inquiry from the Reviewer regarding the selection of the more appropriate technique for insulin content measurement. In this regard, we want to point out that the measurement of pancreatic insulin content performed on whole pancreatic extracts is an accepted approach in many publications (Endocrinol Diabetes Metab. 6,e392, 2023, doi: 10.1002/edm2.392; Environ Sci Pollut Res Int. 20, 2023, doi: 10.1007/s11356-023-31066-7; Nutrients, 1814, 3866, 2022, doi: 10.3390/nu14183866; Mol Nutr Food Res. 59, 820-4, 2015, doi: 10.1002/mnfr.201400746), given that beta cells are the only cells that specifically express and secrete insulin. Indeed, in previous studies we have also isolated islets and measured islet insulin content and glucose-stimulated secretion ex vivo. While this technique is very accurate, its main use is to determine whether changes in insulin secretion are due to defects in the secretory machinery per se or to elucidate whether it is associated with changes in insulin content per islet. The latter was not the aim of the graph included in new Figure 4 (please note that the numbering of the figures has changed in the revised version of the manuscript).

2. Figure 2, 3, and 5 all show data from total pancreas lysates. These data, showing cascade 3 activity, ROS, carbonyl group, GSH, GPX activity, GR activity, and all western blots use total pancreas lysates. This issue inappropriate. Isolated islets should be used.

With the islets being only 2% of the total pancreas weight, the 98% of exocrine pancreas could skew the results. Isolated islets must be used.
We understand the Reviewer's concern, but our main interest in this study was to investigate the potential of CCB supplementation to reverse the oxidative stress and inflammation in the pancreas and thus prevent the loss of functional beta cells in a diabetic context. Therefore, we primarily focused on the pancreas. It is known that increased inflammation and ROS production in the pancreas of T2D patients lead to fibrosis and it consequently affects vascularization and determines whether or not insulin reaches the blood stream efficiently. Moreover, accumulation of fat into the pancreatic parenchyma together with macrophage infiltration negatively impact on beta cell functionality. For these reasons, we focused initially on total pancreatic tissue and markers of stress, inflammation and even apoptosis in whole pancreatic homogenates. However, to assess macrophage infiltration and beta cell damage, we performed a histological study of the pancreas. We fully agree with the Reviewer that the study in isolated islet could have provided complementary information and highlighted the molecular mechanism involved in the beneficial effect of CCB and metformin, but we did not perform this approach because it was not a key point of this study. Besides, it is not possible for us to obtain new animals at this moment to conduct additional experiments. Nevertheless, we strongly believe that the results presented in the manuscript, highlighting the ability of CCB to prevent oxidative stress, inflammation and thus beta cell loss in diabetic animals, as well as their synergistic effect with metformin, are relevant and open the way for future research in this area.
We are fully aware about the importance of the point raised by the reviewer. In response to his/her feedback, and to better clarify our pancreas-focused objective, we have reformatted the presentation of the results and their discussion to follow a more appropriated order as described above. Therefore, in the revised manuscript, we first show markers of stress, inflammation and even apoptosis in whole pancreas homogenates (new Figures 1 and 2). Then we show macrophage infiltration and beta cell damage by histological examination of the pancreas (new Figure 3), and finally we present the effect of this local inflammation on beta cell damage and loss (new Figures 4 and 5). We believe that this reorganisation of the text has significantly improved the comprehensibility of the manuscript.

3. All the graphs should be changed from the plunger plots to bar graphs that show the individual data points.

Following the Reviewer’s advice, all figures have been converted into scatter blot plots to better illustrate individual data points. In addition, as suggested by Referee 2, we have revised the presentation of statistical information. Distinct symbols are now used to denote p-values, with detailed explanations in the figure legends for the specific comparisons.

Reviewer 2 Report

Comments and Suggestions for Authors

The submitted manuscript “Synergistic effect of a flavonoid-rich cocoa-carob blend and metformin in preserving beta cell mass in Zucker diabetic fatty rats” report that feeding of Zucker diabetic fatty rats with CCB supplemented diet ameliorated diabetes by attenuating hyperglycemia, preventing beta cell apoptosis and increasing beta cell mass, decreasing islet inflammation. The submitted manuscript is one of several related to CCB subject from this lab.

General remarks and requests:

The authors should have made it clear that the current study is not a standing alone study but rather a part of the study published  two years ago “Supplementation with a Cocoa–Carob Blend, Alone or in Combination with Metformin, Attenuates Diabetic Cardiomyopathy, Cardiac Oxidative Stress and Inflammation in Zucker Diabetic Rats. Antioxidants 202211(2), 432. The information in Table 1 was simply transferred from the previously published manuscript and this was not acknowledged anywhere in the text, which is not right.

The authors state in Conclusions that the submitted study for the first time demonstrate that cocoa-cabob blend prevents beta cell apoptosis and the loss of beta cell mass, however they cite the papers that have shown just that (one example is Ref#18). The authors must clarify what is new the current study, and clearly state the goal/hypothesis of the study.

Data quality and presentation concerns:

1.       The authors confuse the reader by using a non-conventional presentation of statistics using letters without detailed explanation. They should use stars reflecting p values and lines to specify the comparisons

2.       The graphs should contain single data points in addition to the bar. The number of animals used for the generation of each graph should be stated in the legend

3.       The images of islets do not reflect the data in the corresponding quantification graphs

4.       Results obtained from the whole pancreas either should not be combined with the results obtained from islets in the same figure, or each figure panel should be clearly labeled “islets” or “whole pancreas”, and clear distinction must be made in the text. For example, caspase 3 activity was obtained using the whole pancreas lysate, while TUNEL in islets. Did the authors look at caspase 3 activity in islets using immunostaining?

Interpretation concerns:

1.       The authors in many occasions assign findings from the whole pancreas to islets which is misleading because islets are only 1-2% of total pancreas

2.       The authors measure fractional beta cell area, but talk about beta cell mass and even included “beta cell mass” in the title which is misleading. Fractional area is not an equivalent for beta cell mass. Beta cells mass should be calculated using pancreas weights and presented in the manuscript instead of the insulin positive fractional area

3.       The observed improvement of blood glucose upon treatment most likely not only due to the beta cell preservation, but also at least in part due to the effect of treatments on liver function. This should be addressed in discussion. Did the authors look at insulin signaling properties in livers?

4.       The discussion should contain some thoughts about why metformin sometimes counteracts the positive effect of CCB, and what implications if any it could have for humans since most diabetic patients take metformin

Additional comments:  

-Lines 444-445- the authors state that “CCB diet led to markable recovery of beta cell mass”. They did not measure changes in beta cell mass with time, therefore the use of word “preservation” (relative to untreated rat) is more appropriate

-The discussion should be more specific and primarily focused on the novelty of the results and/or the justification of importance of confirmation of the previously published data in the current study

Comments on the Quality of English Language

here just a few examples where the revision is needed:

-          Line 40- “...beta cells are instrumental in the development and onset of diabetes”. “Instrumental” cannot be used in this context

-          Lines 47, 49, 56, 59, .... need revision and many more

-          Lines 501-502- “CCB supplementation not only normalized the number of pancreatic macrophages, but avoided their infiltration”. I guess the authors wanted to say that the number of macrophages in the pancreas decreased relative to untreated ZDF rats

Author Response

Answer to Reviewer 2

We would like to thank the Reviewer for the careful reading of the manuscript, as well as for the encouraging comments and helpful suggestions. All comments and suggestions have been accepted and we believe that the manuscript has been largely improved by the changes introduced.

General remarks and requests:

1.- The authors should have made it clear that the current study is not a standing alone study but rather a part of the study published two years ago “Supplementation with a Cocoa–Carob Blend, Alone or in Combination with Metformin, Attenuates Diabetic Cardiomyopathy, Cardiac Oxidative Stress and Inflammation in Zucker Diabetic Rats. Antioxidants 2022, 11(2), 432. The information in Table 1 was simply transferred from the previously published manuscript and this was not acknowledged anywhere in the text, which is not right.

We fully appreciate the Reviewer's perspective. This study serves as a continuation of a previously published paper exploring the antidiabetic effects of supplementation with a developed functional food rich in polyphenols, combining cocoa powder and carob flour, focused on the heart in a T2D animal model. While we did allude to this study at the end of the introduction, we acknowledge that it may not have been as evident. Consequently, in this revised version, we have explicitly stated this continuity in both at the end of the introduction section and in materials and methods sections 2.2.

As noted by the Reviewer, Table 1 includes data from the previously published manuscript because it is necessary to understand the biological characteristics of the animals, particularly those related to glucose homeostasis. However, we are aware that these results have been reported previously, so Table 1 is now presented as Supplementary Table 1. We have appropriately indicated that the data are from the previous work and have also asked the Editor to confirm that they can be presented in this form or whether we should seek permission from the Editorial Office.

  1. The authors state in Conclusions that the submitted study for the first time demonstrate that cocoa-cabob blend prevents beta cell apoptosis and the loss of beta cell mass, however they cite the papers that have shown just that (one example is Ref#18). The authors must clarify what is new the current study, and clearly state the goal/hypothesis of the study.

We appreciate the Reviewer's concern, as it appears we may not have sufficiently clarified the novel aspects of our study in articulating its main objectives and conclusions.

In contrast to our previous study, as referred to the reviewer (Ref 18), which focused on a pre-diabetic state in young ZDF rats under normal glycaemic conditions, the current study deal with an established diabetic state in adult ZDF rats with considerably more severe hyperglycaemia. Therefore, the focus on investigating the impact of a potential functional food rich in flavonoids, specifically a blend of cocoa powder and carob flour (CCB), on the preservation of pancreatic beta cells in the context of established T2D stands out as a significant and novel aspect of our research. Moreover, the exploration of potential synergistic effects arising from the combination of this dietary supplementation with metformin, a keystone in antidiabetic drug therapy, introduces an additional layer of novelty to our study. We firmly believe that the main conclusions of our current study, highlighting the capacity of CCB to prevent oxidative stress and inflammation in the pancreas of diabetic animals, ultimately preventing beta cell apoptosis and the loss of functional beta cells in ZDF diabetic animals, along with the observed additive effect when CCB is combining with metformin, represent a substantial advance in the understanding of new potential therapeutic approaches for T2D.

Following the Reviewer's recommendations, in the revised version, we have more clearly articulated the objective of the study both in the abstract and at the end of the introduction section.

Data quality and presentation concerns:

  1. The authors confuse the reader by using a non-conventional presentation of statistics using letters without detailed explanation. They should use stars reflecting p values and lines to specify the comparisons.

In response to the Referee's guidance, we have made adjustments to improve the presentation of statistical data. We have introduced different symbols for the presentation of p-values, with detailed explanations in the figure legends for the specific comparisons. These changes aim to improve the clarity and interpretation of the statistical analyses of our study, without losing any of the essential information.

  1. The graphs should contain single data points in addition to the bar. The number of animals used for the generation of each graph should be stated in the legend.

In line with the Reviewer's suggestion, we have revised all figures in the manuscript, presenting them as scatter dot blot bar graphs. Additionally, we have included information in the figure legends specifying the number of animals used.

  1. The images of islets do not reflect the data in the corresponding quantification graphs

We are not sure about the discordance referred to by the Reviewer between the histology images and the quantification of the graphs as we have revised the manuscript in detail, and we strongly believe that the images chosen accurately represent the results shown. We would be grateful if the reviewer could kindly indicate which figure and section, he/she is referring to so that we can modify it appropriately to meet his/her expectations.

  1. Results obtained from the whole pancreas either should not be combined with the results obtained from islets in the same figure, or each figure panel should be clearly labeled “islets” or “whole pancreas”, and clear distinction must be made in the text. For example, caspase 3 activity was obtained using the whole pancreas lysate, while TUNEL in islets. Did the authors look at caspase 3 activity in islets using immunostaining?

We understand the concern of the Reviewer. In response to his/her comments, and also in agreement to suggestions of Reviewer 1, we have changed the presentation of the results to follow a more appropriated order and to avoid presenting the results obtained from pancreatic homogenates with the results obtained from islets in the same figure. Therefore, in the revised manuscript, we first show the pro-oxidant and inflammatory environment in the pancreas by evaluating markers of stress, inflammation and even apoptosis (caspase-3 activity) in whole pancreatic homogenates (new Figures 1 and 2). Since this diabetic environment leads to an increase in the number of macrophages and their infiltration of the islets, we then show macrophage infiltration and beta cell damage by histological examination of the pancreas (new Figure 3). Finally, we show the effect of this local inflammation on beta cell damage and loss (new Figures 4 and 5). We believe that this reorganisation of the text has greatly improved the readability of the manuscript and avoids potential confusion. However, with regard to the pancreatic insulin content, we have kept the pancreas data together with the relative beta cell mass data, because measuring pancreatic insulin content from whole pancreas extracts is a widely accepted approach to reflect the islet insulin content. As beta cells are the only cells that specifically express and secrete insulin, this method provides valuable information.

Interpretation concerns:

  1. The authors in many occasions assign findings from the whole pancreas to islets which is misleading because islets are only 1-2% of total pancreas

We fully agree with this consideration. As mentioned above, we have addressed this concern by making changes to the presentation of the study results. In addition, we have revised the entire manuscript to avoid this error.

  1. The authors measure fractional beta cell area but talk about beta cell mass and even included “beta cell mass” in the title which is misleading. Fractional area is not an equivalent for beta cell mass. Beta cells mass should be calculated using pancreas weights and presented in the manuscript instead of the insulin positive fractional area

We would like to apologise for the error of using 'beta cell mass' instead of the correct term 'beta cell fractional area' in the original manuscript. We have thoroughly reviewed the entire manuscript and corrected this error, ensuring that the accurate terminology is used in the revised version. In addition, the title of the manuscript has been changed accordingly.

  1. The observed improvement of blood glucose upon treatment most likely not only due to the beta cell preservation, but also at least in part due to the effect of treatments on liver function. This should be addressed in discussion. Did the authors look at insulin signalling properties in livers?

This is an interesting point brought out by the Reviewer. Actually, as showed in Table 1 (HOMA IR values), ZDF animals also exhibited an increase in insulin resistance, a condition that was significantly reduced by both CCB supplementation and metformin treatment. Unfortunately, we have not specifically evaluated the effect of CCB on the liver function in ZDF rats, but our previous results indicate that cocoa flavanols, under insulin resistance conditions, can exert an insulin-like effect on human hepatic HepG2 cells. This effect involves attenuating the blockade of the insulin signaling cascade and modulating glucose uptake and production (Food Chem. Toxicol. 2014, 64, 10–19). Additionally, in pre-diabetic rats, a cocoa-enriched diet demonstrated the ability to alleviate hepatic insulin resistance by regulating key proteins of the insulin pathway and glucose metabolism in the liver (Journal of Nutritional Biochemistry, 26: 704–712, 2015). Therefore, it cannot be ruled out that the observed antidiabetic properties of CCB supplementation might be partly attributed to flavanols' effects on the liver function and insulin resistance, hallmarks of T2D development and progression.

We are perfectly aware about the relevance of the issue and thus it has been duly addressed adding a new paragraph on the topic. This concern has been now included at the end of the discussion in the new revised version.

  1. The discussion should contain some thoughts about why metformin sometimes counteracts the positive effect of CCB, and what implications if any it could have for humans since most diabetic patients take metformin

Although metformin is the first-line treatment for T2D, whether it reverses islet function defects in fully established T2D human patients is an open question. Indeed, the direct effect of metformin on beta-cell function is controversial. Some studies have reported that long-term treatment with metformin of islets from T2D donors partially restored islet insulin content and glucose stimulated insulin secretion as well as reduced oxidative stress and apoptosis (J. Clin. Endocrinol. Metab. 2004, 89, 5535–5541). However, others authors have described inhibition of insulin secretion from both human and rodent islets (Am J Physiol Endocrinol Metab. 2004 Jun;286(6): E1023-31. doi: 10.1152/ajpendo.00532.2003; J. Diabetes Res. 2018, 2018, 9163052) and beta-cells (Am J Physiol Endocrinol Metab. 2004 Jun;286(6): E1023-31. doi: 10.1152/ajpendo.00532.2003) or even no effect (Diabetes Res. Clin. Pract. 2014, 104, 163–170.). In addition, and related to the islet function, Tajima et al. found that metformin abolished beta-cell mass expansion in obesity-induced prediabetic mice together with a marked reduction in insulin secretion (Am. J. Physiol. Endocrinol. Metab. 2017, 313, E367–E380) whereas in prediabetic Nail rats metformin did not impact beta-cell compensation (Int. J. Mol. Sci. 2021, 22, 421. doi: 10.3390/ijms22010421). Although some in vitro evidences have shown that metformin suppresses proliferation and induces apoptosis, the decreased cleaved caspase 3 observed by us and others (Int. J. Mol. Sci. 2021, 22, 421. doi: 10.3390/ijms22010421), as well as the recovery of beta-cell mass, indicated no proapoptotic activity of metformin.

Despite the pharmacological action of metformin is classically through an AMPK-dependent mechanism, the reduction of fibrosis induced by TGFβ after metformin treatment has been described as an AMPK-independent process in salivary human glands (Int J Mol Sci. 2023 Nov 13;24(22):16260. doi: 10.3390/ijms242216260). Since flavonoid-derived polyphenols may also exert their beneficial effects through AMPK-dependent and independent mechanisms of action (J Agric Food Chem. 2023, 71, 17554-17569), we cannot rule out that metformin may interfere at some molecular level with the antifibrotic effect of the CCB diet in ZDF rats when both treatments are administered together, counteracting the former.

Taken together, and despite the conflicting role indicated by the literature, metformin treatment, alone or in combination with the CCB diet, does not appear to exert a global negative effect on beta-cell function or survival, but instead metformin may help preserve the beta-cell mass in a diabetic environment by reducing the pancreatic oxidative and inflammatory profile.

Following the Referee's suggestion, the Discussion has been extended with a paragraph discussing the apparently controversial results obtained with metformin.

Additional comments: 

  1. Lines 444-445- the authors state that “CCB diet led to markable recovery of beta cell mass”. They did not measure changes in beta cell mass with time, therefore the use of word “preservation” (relative to untreated rat) is more appropriate

The sentence has been corrected following the Reviewer indication.

  1. The discussion should be more specific and primarily focused on the novelty of the results and/or the justification of importance of confirmation of the previously published data in the current study

As indicated by the Reviewer, and with the reorganisation of the results, the discussion has been revised to make it more focused on the objectives and novelty of the study.

Comments on the Quality of English Language

  1. Line 40- “...beta cells are instrumental in the development and onset of diabetes”. “Instrumental” cannot be used in this context

According to the Reviewer's indication, we have replaced the word "instrumental" in the sentence

  1. Lines 47, 49, 56, 59, …. need revision and many more

We have corrected the sentences indicated by the Reviewer, and we have double-checked the language throughout the revised manuscript.

  1. Lines 501-502- “CCB supplementation not only normalized the number of pancreatic macrophages, but avoided their infiltration”. I guess the authors wanted to say that the number of macrophages in the pancreas decreased relative to untreated ZDF rats

According to the Reviewer, we have rewritten this sentence in the text.

Round 2

Reviewer 1 Report

Comments and Suggestions for Authors

Please describe the whole pancreas approach as a limitation of the study as it does not allow you to directly tie the biochemical changes to the islets, but rather the entire pancreas.

Author Response

Reviewer 1

Please describe the whole pancreas approach as a limitation of the study as it does not allow you to directly tie the biochemical changes to the islets, but rather the entire pancreas.

Following the Reviewer's feedback, we have acknowledged at the end of the Discussion section that our study primarily focused on the effects in the whole pancreas, which is a limitation as it limits our ability to directly correlate the functional changes in the islets. Future studies using isolated islets may provide additional insights and elucidate the molecular mechanisms underlying the beneficial effects of CCBs and metformin on beta cells.

Reviewer 2 Report

Comments and Suggestions for Authors

Answer to Reviewer 2

We would like to thank the Reviewer for the careful reading of the manuscript, as well as for the encouraging comments and helpful suggestions. All comments and suggestions have been accepted and we believe that the manuscript has been largely improved by the changes introduced.

Reviewer: The authors significantly improved the manuscript by re-arranging the order of the narrative, updating figures and figure legends. However, multiple shortcomings in data analysis as well as writing remain and must be addressed. Details are below the authors responses.

General remarks and requests:

1.- The authors should have made it clear that the current study is not a standing alone study but rather a part of the study published two years ago “Supplementation with a Cocoa–Carob Blend, Alone or in Combination with Metformin, Attenuates Diabetic Cardiomyopathy, Cardiac Oxidative Stress and Inflammation in Zucker Diabetic Rats. Antioxidants 2022, 11(2), 432. The information in Table 1 was simply transferred from the previously published manuscript and this was not acknowledged anywhere in the text, which is not right.

We fully appreciate the Reviewer's perspective. This study serves as a continuation of a previously published paper exploring the antidiabetic effects of supplementation with a developed functional food rich in polyphenols, combining cocoa powder and carob flour, focused on the heart in a T2D animal model. While we did allude to this study at the end of the introduction, we acknowledge that it may not have been as evident. Consequently, in this revised version, we have explicitly stated this continuity in both at the end of the introduction section and in materials and methods sections 2.2.

As noted by the Reviewer, Table 1 includes data from the previously published manuscript because it is necessary to understand the biological characteristics of the animals, particularly those related to glucose homeostasis. However, we are aware that these results have been reported previously, so Table 1 is now presented as Supplementary Table 1. We have appropriately indicated that the data are from the previous work and have also asked the Editor to confirm that they can be presented in this form or whether we should seek permission from the Editorial Office.

Reviewer: Thank you for the mentioning continuity of the study and moving the table to the supplement.

Table should contain body weights.

  1. The authors state in Conclusions that the submitted study for the first time demonstrate that cocoa-cabob blend prevents beta cell apoptosis and the loss of beta cell mass, however they cite the papers that have shown just that (one example is Ref#18). The authors must clarify what is new the current study, and clearly state the goal/hypothesis of the study.

We appreciate the Reviewer's concern, as it appears we may not have sufficiently clarified the novel aspects of our study in articulating its main objectives and conclusions.

In contrast to our previous study, as referred to the reviewer (Ref 18), which focused on a pre-diabetic state in young ZDF rats under normal glycaemic conditions, the current study deal with an established diabetic state in adult ZDF rats with considerably more severe hyperglycaemia. Therefore, the focus on investigating the impact of a potential functional food rich in flavonoids, specifically a blend of cocoa powder and carob flour (CCB), on the preservation of pancreatic beta cells in the context of established T2D stands out as a significant and novel aspect of our research. Moreover, the exploration of potential synergistic effects arising from the combination of this dietary supplementation with metformin, a keystone in antidiabetic drug therapy, introduces an additional layer of novelty to our study. We firmly believe that the main conclusions of our current study, highlighting the capacity of CCB to prevent oxidative stress and inflammation in the pancreas of diabetic animals, ultimately preventing beta cell apoptosis and the loss of functional beta cells in ZDF diabetic animals, along with the observed additive effect when CCB is combining with metformin, represent a substantial advance in the understanding of new potential therapeutic approaches for T2D.

Following the Reviewer's recommendations, in the revised version, we have more clearly articulated the objective of the study both in the abstract and at the end of the introduction section.

Reviewer: it would be beneficial mentioning in line 76:  “ ... and in vivo in young prediabetic rats [18].”

Data quality and presentation concerns:

  1. The authors confuse the reader by using a non-conventional presentation of statistics using letters without detailed explanation. They should use stars reflecting p values and lines to specify the comparisons.

In response to the Referee's guidance, we have made adjustments to improve the presentation of statistical data. We have introduced different symbols for the presentation of p-values, with detailed explanations in the figure legends for the specific comparisons. These changes aim to improve the clarity and interpretation of the statistical analyses of our study, without losing any of the essential information.

Reviewer: The authors significantly improved data presentation by clarifying the number of animals per group and introducing symbols for statistics together with the description of comparisons. In line 256 they described SEM as standard error of the median; did they intended to write standard error of the mean?

 The graphs should contain single data points in addition to the bar. The number of animals used for the generation of each graph should be stated in the legend.

In line with the Reviewer's suggestion, we have revised all figures in the manuscript, presenting them as scatter dot blot bar graphs. Additionally, we have included information in the figure legends specifying the number of animals used.

 Reviewer: Thank you, it is helpful.

  1. The images of islets do not reflect the data in the corresponding quantification graphs

We are not sure about the discordance referred to by the Reviewer between the histology images and the quantification of the graphs as we have revised the manuscript in detail, and we strongly believe that the images chosen accurately represent the results shown. We would be grateful if the reviewer could kindly indicate which figure and section, he/she is referring to so that we can modify it appropriately to meet his/her expectations.

Reviewer: Figure3a:  ZDF (CCB) panel has large islet implying the increase in islet size. Is this a representative islet really? I am sure the authors could use a different more representative image. How the boundaries of the islets were defined given the disrupted islet morphology? The authors should specify which macrophages they considered to be peripheral, and describe the quantification approach in the method section. In the figure panels, the authors should identify peripheral macrophages and exocrine tissue macrophages using different symbols. The figure legend should contain description, for example: star – intra-islet macrophages, closed arrowhead – peripheral, and open arrowhead – exocrine).

  1. Results obtained from the whole pancreas either should not be combined with the results obtained from islets in the same figure, or each figure panel should be clearly labeled “islets” or “whole pancreas”, and clear distinction must be made in the text. For example, caspase 3 activity was obtained using the whole pancreas lysate, while TUNEL in islets. Did the authors look at caspase 3 activity in islets using immunostaining?

We understand the concern of the Reviewer. In response to his/her comments, and also in agreement to suggestions of Reviewer 1, we have changed the presentation of the results to follow a more appropriated order and to avoid presenting the results obtained from pancreatic homogenates with the results obtained from islets in the same figure. Therefore, in the revised manuscript, we first show the pro-oxidant and inflammatory environment in the pancreas by evaluating markers of stress, inflammation and even apoptosis (caspase-3 activity) in whole pancreatic homogenates (new Figures 1 and 2). Since this diabetic environment leads to an increase in the number of macrophages and their infiltration of the islets, we then show macrophage infiltration and beta cell damage by histological examination of the pancreas (new Figure 3). Finally, we show the effect of this local inflammation on beta cell damage and loss (new Figures 4 and 5). We believe that this reorganisation of the text has greatly improved the readability of the manuscript and avoids potential confusion. However, with regard to the pancreatic insulin content, we have kept the pancreas data together with the relative beta cell mass data, because measuring pancreatic insulin content from whole pancreas extracts is a widely accepted approach to reflect the islet insulin content. As beta cells are the only cells that specifically express and secrete insulin, this method provides valuable information.

Reviewer: What is “the relative beta cell mass” mentioned above? The beta cell mass data was not included in the paper, while it is very relevant and necessary. The authors are reluctant to present

calculated beta cell mass, why? Was the pancreatic weight different between the groups? 

Interpretation concerns:

  1. The authors in many occasions assign findings from the whole pancreas to islets which is misleading because islets are only 1-2% of total pancreas

We fully agree with this consideration. As mentioned above, we have addressed this concern by making changes to the presentation of the study results. In addition, we have revised the entire manuscript to avoid this error. 

  1. The authors measure fractional beta cell area but talk about beta cell mass and even included “beta cell mass” in the title which is misleading. Fractional area is not an equivalent for beta cell mass. Beta cells mass should be calculated using pancreas weights and presented in the manuscript instead of the insulin positive fractional area

We would like to apologise for the error of using 'beta cell mass' instead of the correct term 'beta cell fractional area' in the original manuscript. We have thoroughly reviewed the entire manuscript and corrected this error, ensuring that the accurate terminology is used in the revised version. In addition, the title of the manuscript has been changed accordingly.

Reviewer: Thank you for the clarification. Line 357 still states “beta cell mass” however.

The authors show an increase in insulin content upon treatment with CCB, an increase in fractional beta cell area and decrease in apoptosis. However, all three do not provide the evidence of the direct effect of treatment on beta cell preservation. There could be two things happening: the improvement in insulin sensitivity results in less secretion, less stress and therefore an increase in insulin content; or/and TUNEL decreases because there is less turnover due to less stress on beta cells due to decrease in demand for secretion. Did the authors look whether beta cell proliferation is influenced by treatments?

  1. The observed improvement of blood glucose upon treatment most likely not only due to the beta cell preservation, but also at least in part due to the effect of treatments on liver function. This should be addressed in discussion. Did the authors look at insulin signalling properties in livers?

This is an interesting point brought out by the Reviewer. Actually, as showed in Table 1 (HOMA IR values), ZDF animals also exhibited an increase in insulin resistance, a condition that was significantly reduced by both CCB supplementation and metformin treatment. Unfortunately, we have not specifically evaluated the effect of CCB on the liver function in ZDF rats, but our previous results indicate that cocoa flavanols, under insulin resistance conditions, can exert an insulin-like effect on human hepatic HepG2 cells. This effect involves attenuating the blockade of the insulin signaling cascade and modulating glucose uptake and production (Food Chem. Toxicol. 2014, 64, 10–19). Additionally, in pre-diabetic rats, a cocoa-enriched diet demonstrated the ability to alleviate hepatic insulin resistance by regulating key proteins of the insulin pathway and glucose metabolism in the liver (Journal of Nutritional Biochemistry, 26: 704–712, 2015). Therefore, it cannot be ruled out that the observed antidiabetic properties of CCB supplementation might be partly attributed to flavanols' effects on the liver function and insulin resistance, hallmarks of T2D development and progression.

We are perfectly aware about the relevance of the issue and thus it has been duly addressed adding a new paragraph on the topic. This concern has been now included at the end of the discussion in the new revised version.

 Reviewer: the added paragraph is relevant.

  1. The discussion should contain some thoughts about why metformin sometimes counteracts the positive effect of CCB, and what implications if any it could have for humans since most diabetic patients take metformin

Although metformin is the first-line treatment for T2D, whether it reverses islet function defects in fully established T2D human patients is an open question. Indeed, the direct effect of metformin on beta-cell function is controversial. Some studies have reported that long-term treatment with metformin of islets from T2D donors partially restored islet insulin content and glucose stimulated insulin secretion as well as reduced oxidative stress and apoptosis (J. Clin. Endocrinol. Metab. 2004, 89, 5535–5541). However, others authors have described inhibition of insulin secretion from both human and rodent islets (Am J Physiol Endocrinol Metab. 2004 Jun;286(6): E1023-31. doi: 10.1152/ajpendo.00532.2003; J. Diabetes Res. 2018, 2018, 9163052) and beta-cells (Am J Physiol Endocrinol Metab. 2004 Jun;286(6): E1023-31. doi: 10.1152/ajpendo.00532.2003) or even no effect (Diabetes Res. Clin. Pract. 2014, 104, 163–170.). In addition, and related to the islet function, Tajima et al. found that metformin abolished beta-cell mass expansion in obesity-induced prediabetic mice together with a marked reduction in insulin secretion (Am. J. Physiol. Endocrinol. Metab. 2017, 313, E367–E380) whereas in prediabetic Nail rats metformin did not impact beta-cell compensation (Int. J. Mol. Sci. 2021, 22, 421. doi: 10.3390/ijms22010421). Although some in vitro evidences have shown that metformin suppresses proliferation and induces apoptosis, the decreased cleaved caspase 3 observed by us and others (Int. J. Mol. Sci. 2021, 22, 421. doi: 10.3390/ijms22010421), as well as the recovery of beta-cell mass, indicated no proapoptotic activity of metformin.

Despite the pharmacological action of metformin is classically through an AMPK-dependent mechanism, the reduction of fibrosis induced by TGFβ after metformin treatment has been described as an AMPK-independent process in salivary human glands (Int J Mol Sci. 2023 Nov 13;24(22):16260. doi: 10.3390/ijms242216260). Since flavonoid-derived polyphenols may also exert their beneficial effects through AMPK-dependent and independent mechanisms of action (J Agric Food Chem. 2023, 71, 17554-17569), we cannot rule out that metformin may interfere at some molecular level with the antifibrotic effect of the CCB diet in ZDF rats when both treatments are administered together, counteracting the former.

Taken together, and despite the conflicting role indicated by the literature, metformin treatment, alone or in combination with the CCB diet, does not appear to exert a global negative effect on beta-cell function or survival, but instead metformin may help preserve the beta-cell mass in a diabetic environment by reducing the pancreatic oxidative and inflammatory profile.

Following the Referee's suggestion, the Discussion has been extended with a paragraph discussing the apparently controversial results obtained with metformin.

Reviewer: The extension of the discussion was not asked for and listing information was not helpful and rather irrelevant. Just say you do not know why.

Additional comments: 

  1. Lines 444-445- the authors state that “CCB diet led to markable recovery of beta cell mass”. They did not measure changes in beta cell mass with time, therefore the use of word “preservation” (relative to untreated rat) is more appropriate

The sentence has been corrected following the Reviewer indication.

  1. The discussion should be more specific and primarily focused on the novelty of the results and/or the justification of importance of confirmation of the previously published data in the current study

As indicated by the Reviewer, and with the reorganisation of the results, the discussion has been revised to make it more focused on the objectives and novelty of the study.

Reviewer: The discussion is too long and lacks clarity, should be revised.

There are multiple occasions of repetitions like lines 514 and 518 about insulin content.

Lines 518-522 should be removed altogether because the authors did not assess islet size or neogenesis; small clusters of beta cells could be parts of islets with disrupted morphology as it appears on the images in figure 4a rather then “initiation of neogenesis processes”. It is not clear what “initiation’ even mean in this context.

I would suggest to focus in short on the following:

CCB treatment results in decrease in pancreas inflammation and an increase in insulin sensitivity as expected based on the previously reported anti-inflammatory properties of flavonoids and positive effects on insulin sensitivity. This is confirmatory part of the paper.

New findings:

On the level of the whole pancreas an addition of CCB to metformin might be beneficial since it further reduces levels of proinflammatory cytokines and reduces number of macrophages, and also has mild positive effect on oxidative stress (further increases in GSH and GPx activity without an additional effect on the other readout of oxidative stress) all these are relevant to pancreatitis which is a known diabetic complication.

On the level of the islet: while metformin treatment does not have an effect on the number of macrophages in the whole pancreas, it mitigates islet infiltration, and the addition of CCB amplifies this effect, which could be beneficial for beta cell function.

Comments on the Quality of English Language

  1. Line 40- “...beta cells are instrumental in the development and onset of diabetes”. “Instrumental” cannot be used in this context

According to the Reviewer's indication, we have replaced the word "instrumental" in the sentence

Reviewer: I would suggest removing lines 41-46, they are repetitive and not needed.

  1. Lines 47, 49, 56, 59, …. need revision and many more

We have corrected the sentences indicated by the Reviewer, and we have double-checked the language throughout the revised manuscript.

Reviewer: The words “critical, severe, significantly” in lines 50-57 have no scientific meaning in this context, and could be easily removed.

Lines 58-60: The sentence is meaningless, adds no value and should be removed.

Lines 395: “...all treatments tested resulted in a significantly larger fractional beta cell area...”

Of note, this sentence may change depending on the results of beta cell mass calculation which the authors will hopefully provide.

Lines 401-406: I suggest changing to “...ZDF rats (Figure 4d). ” Delete lines 401-403.

Section 3.4 either should be re-named (it is not clear what “damage” means here) or may be combined with 3.5.

Lines 501-502- “CCB supplementation not only normalized the number of pancreatic macrophages, but avoided their infiltration”. I guess the authors wanted to say that the number of macrophages in the pancreas decreased relative to untreated ZDF rats

Comments on the Quality of English Language

The narrative should be shorten to avoid repetitions and usage of words that have no scientific meaning.  

Author Response

We wish to thank the Reviewer for the careful reading of the manuscript, as well as for their compliments and interesting comments.

  1. Table should contain body weights.

Following the Reviewer's suggestions, we have added the body weight in the supplementary Table 1.

  1. Reviewer: it would be beneficial mentioning in line 76:  “ ... and in vivo in young prediabetic rats [18].”

Accordingly, we have now indicated that the in vivo study was conducted in young prediabetic rats.

  1. The authors significantly improved data presentation by clarifying the number of animals per group and introducing symbols for statistics together with the description of comparisons. In line 256 they described SEM as standard error of the median; did they intended to write standard error of the mean?

Indeed, it was a mistake. We meant to indicate the standard error of the mean. It has been corrected in the text.

  1. Figure3a:  ZDF (CCB) panel has large islet implying the increase in islet size. Is this a representative islet really? I am sure the authors could use a different more representative image. How the boundaries of the islets were defined given the disrupted islet morphology? The authors should specify which macrophages they considered to be peripheral, and describe the quantification approach in the method section. In the figure panels, the authors should identify peripheral macrophages and exocrine tissue macrophages using different symbols. The figure legend should contain description, for example: star – intra-islet macrophages, closed arrowhead – peripheral, and open arrowhead – exocrine).

As suggested by Reviewer, the corresponding image in Figure 3a has been replaced by another to more accurately reflect the results obtained. In this study, we found two different types of pancreatic islets in the ZDF(CCB) state: some were large islets with strong insulin staining that did not completely recover the classic round shape of lean islets, and the others were islets of similar size and shape to lean islets. For this reason, the previous image may be confusing.

Despite disrupted islet morphology in diabetic animals, especially in non-treated ZDF rats, fibrotic tissue was frequently localised in the peripheral islet area and even more frequently in close proximity to the vessels, but not inside the islet, which allowed us to identify the border area of the islet using the micrometre of the microscope.

Following the reviewer's suggestion, the Materials and Methods have been rewritten to specify the criteria used for CD68 cell counting. The description of macrophages has been added to the figure legend.

  1. Reviewer: What is “the relative beta cell mass” mentioned above? The beta cell mass data was not included in the paper, while it is very relevant and necessary. The authors are reluctant to present calculated beta cell mass, why? Was the pancreatic weight different between the groups? 

We fully agree with the Reviewer that beta cell mass data would be relevant in this study given the difference in the body weight between ZL and ZDF animals. Unfortunately, we do not have the pancreas weights of the different animals to calculate the beta cell mass. Nevertheless, we believe that the beta cell fraction results allow us to elucidate important and meaningful information regarding the effect (direct or indirect) that the CCB diet, alone or in combination with metformin, may have in the context of diabetes.

We apologize by the mistake using the term ‘relative beta cell mass’ as we indeed referred to beta cell fraction.

  1. Thank you for the clarification. Line 357 still states “beta cell mass” however.

The authors show an increase in insulin content upon treatment with CCB, an increase in fractional beta cell area and decrease in apoptosis. However, all three do not provide the evidence of the direct effect of treatment on beta cell preservation. There could be two things happening: the improvement in insulin sensitivity results in less secretion, less stress and therefore an increase in insulin content; or/and TUNEL decreases because there is less turnover due to less stress on beta cells due to decrease in demand for secretion. Did the authors look whether beta cell proliferation is influenced by treatments?

We fully agree with the Reviewer. It is true that with the results of this work we cannot affirm that the observed effect on beta cells is due to a direct effect of CCB. On the contrary, as the reviewer suggests, it may also be due to the effects of CCB on insulin resistance and reduced insulin demand. It is even more likely that the observed effect is due to the combination of both direct and indirect actions. On the other hand, unfortunately, we have not measured beta cell proliferation in this study, which may also be involved in the observed effects on beta cells.

  1. The extension of the discussion was not asked for and listing information was not helpful and rather irrelevant. Just say you do not know why

In agreement with the reviewer, we have removed the specified information and indicated that we have no explanation for it.

  1. 8. The discussion is too long and lacks clarity, should be revised.

There are multiple occasions of repetitions like lines 514 and 518 about insulin content. Lines 518-522 should be removed altogether because the authors did not assess islet size or neogenesis; small clusters of beta cells could be parts of islets with disrupted morphology as it appears on the images in figure 4a rather than “initiation of neogenesis processes”. It is not clear what “initiation’ even mean in this context.

I would suggest to focus in short on the following: CCB treatment results in decrease in pancreas inflammation and an increase in insulin sensitivity as expected based on the previously reported anti-inflammatory properties of flavonoids and positive effects on insulin sensitivity. This is confirmatory part of the paper.

New findings: On the level of the whole pancreas an addition of CCB to metformin might be beneficial since it further reduces levels of proinflammatory cytokines and reduces number of macrophages, and also has mild positive effect on oxidative stress (further increases in GSH and GPx activity without an additional effect on the other readout of oxidative stress) all these are relevant to pancreatitis which is a known diabetic complication. On the level of the islet: while metformin treatment does not have an effect on the number of macrophages in the whole pancreas, it mitigates islet infiltration, and the addition of CCB amplifies this effect, which could be beneficial for beta cell function.

We fully agree with the reviewer and have therefore removed the lines 514-522. In addition, we have shortened the discussion and focused it on what is really important, following the reviewer's indications.

  1. I would suggest removing lines 41-46, they are repetitive and not needed.

The lines 41-46 have been removed.

  1. The words “critical, severe, significantly” in lines 50-57 have no scientific meaning in this context, and could be easily removed.

The words have been removed

  1. Lines 58-60: The sentence is meaningless, adds no value and should be removed.

In agreement with the Reviewer, we have deleted lines 58-60.

  1. Lines 395: “...all treatments tested resulted in a significantly larger fractional beta cell area...”

Of note, this sentence may change depending on the results of beta cell mass calculation which the authors will hopefully provide.

Unfortunately, as indicated below, we do not have the data for beta cell mass, so the sentence has not been changed.

  1. Lines 401-406: I suggest changing to “...ZDF rats (Figure 4d).” Delete lines 401-403.

The changes suggested by the Reviewer have been included in the revised version 2.

  1. Section 3.4 either should be re-named (it is not clear what “damage” means here) or may be combined with 3.5.

In agreement with the reviewer, we have merged sections 3.4 and 3.5 into a single section 3.4.

  1. The narrative should be shorten to avoid repetitions and usage of words that have no scientific meaning.

Following the indications of the Reviewer, we have revised the whole document to try to shorten it avoiding repetitions and the use of words without scientific meaning.